# *Ailanthus altissima* Seed Oil—A Valuable Source of Lipid-Soluble Components with DNA Protective and Antiproliferative Activities

**DOI:** 10.3390/foods13081268

**Published:** 2024-04-21

**Authors:** Tsvetelina Andonova, Zhana Petkova, Olga Teneva, Ginka Antova, Elena Apostolova, Samir Naimov, Tsvetelina Mladenova, Iliya Slavov, Ivanka Dimitrova-Dyulgerova

**Affiliations:** 1Department of Botany and Biological Education, Faculty of Biology, University of Plovdiv “Paisii Hilendarski”, 24 Tzar Asen Street, 4000 Plovdiv, Bulgaria; ts_andonova@uni-plovdiv.bg (T.A.); cmladenova@uni-plovdiv.bg (T.M.); ivadim@uni-plovdiv.bg (I.D.-D.); 2Department of Chemical Technology, University of Plovdiv “Paisii Hilendarski”, 24 Tzar Asen Street, 4000 Plovdiv, Bulgaria; olga@uni-plovdiv.bg (O.T.); ginant@uni-plovdiv.bg (G.A.); 3Department of Plant Physiology and Molecular Biology, Faculty of Biology, University of Plovdiv “Paisii Hilendarski”, 24 Tzar Asen Street, 4000 Plovdiv, Bulgaria; eapostolova@uni-plovdiv.bg (E.A.); naimov0@uni-plovdiv.bg (S.N.); 4Department of Biology, Faculty of Pharmacy, Medical University of Varna, 9000 Varna, Bulgaria; ijelev80@abv.bg

**Keywords:** *Ailanthus altissima*, seed oil, GC compound identification, physicochemical properties, antiproliferative activity, DNA protection

## Abstract

The present study is focused on the chemical and lipid composition of seed oil of the European ornamental and invasive wood plant *Ailanthus altissima* (Simaroubaceae). Total lipids, proteins, carbohydrates, ash, and moisture in the seeds were determined. A high yield of glyceride oil (30.7%) was found, as well as a high content of fibers (29.6%) and proteins (18.7%). Physicochemical properties of the oil define it as semi-dry (129.4 g I_2_/100 g Iodine value) with oxidative stability, refractive index, saponification value, and relative density similar to widely used oils with nutritional value and health benefits. The composition of the seed oil was determined chromatographically. Unsaturated fatty acids (95.3%) predominated in the seed oil, of which linoleic acid (48.6%) and oleic acid (44.8%) were the major ones. The main lipid-soluble bioactive components were *β*-sitosterol (72.6%), *γ*-tocopherol (74.6%), phosphatidylinositol (29.5%), and phosphatidic acids (25.7%). The proven in vitro DNA-protective ability of seed oil is reported for the first time. The seed oil exhibited a weak antiproliferative effect on HT-29 and PC3 tumor cell lines and showed no cytotoxicity on the BALB/c 3T3 cell line. In brief, the present study reveals that *A. altissima* seed oil can be used as a healthy food.

## 1. Introduction

The beneficial effect of vegetable oils on human health is the reason for the growing interest in them. Bringing clarity to their component composition will assist consumers in making the right and informed choices to meet their individual needs [1]. Oilseeds are a source of bioactive components (tocopherols, phenols, carotenoids, phytosterols, phospholipids, etc.) and fatty acids that are part of a balanced and healthy human diet, which is a major factor in human health [2]. A number of their biological activities have been proven, such as anti-inflammatory, antidiabetic, antioxidant, prevention of cardiovascular and cancer diseases, protection of lungs, kidneys, and liver, treatment of diseases such as metabolic and premenstrual syndrome, menopause, diabetes, etc. [1,2]. Seeds of widely used oil-bearing and medicinal plants are mostly reported as sources of beneficial lipids. Other reliable sources of healthy unsaturated fatty acids and fat-soluble phytochemicals are also being sought, such as the oil obtained from the seeds of the woody species *Ailanthus altissima* (Mill.) Swingle [3]. The widespread tree is of Asian origin (China, Taiwan, India), with commonly known names such as tree of heaven, ailanthus, paradise tree, Chinese sumac, etc. It is an invasive alien plant in Europe and North America, distinguished by higher fertility compared to the native species [4,5,6,7]. *Ailanthus* produces large quantities of fruits (samaras) of 300,000 to over one million per year that are wind-dispersed over a long distance (about 100 m) [8,9,10]. For a successful conquest of new territories and displacement of native flora, its undemanding nature to environmental conditions, rapid growth, powerful root system, and secretion of ailanthone, a compound suppressing surrounding vegetation, also play a role [5,10].

The fruits are a source of a wide range of bioactive chemical constituents that are the basis of their use in traditional Chinese medicine, ground into powder, as an anti-dysentery, anti-ascariasis, and astringent agent [11,12]. Ethanol and methanol extracts of ailanthus fruits and sterol compounds isolated from them exhibit in vitro antibacterial (against *Pseudomonas aeruginosa*, *Escherichia coli*, *Staphylococcus aureus*, and *Salmonella typhimurium*), antifungal (against *Microsporum canis*), insecticidal, cytotoxic, and phytotoxic activities [11,12,13]. Inhibitory activity against the Tobacco mosaic virus (TMV) is reported for phenolic derivatives (flavonoids, coumarins, lignans, neolignans, quassinoids, etc.) in the fruit methanol extracts [14,15,16,17]. The ability to scavenge free radicals and reduce oxidative stress are also among the activities analyzed for fruit extracts [18].

Data on the component composition and biological properties exhibited by the fruits are mostly based on their extracts, while information on the physicochemical characteristics, chemical composition, and biological effects of their seed oil is incomplete. Bory and Clair-Maezulajtys reported that phospholipids (phosphatidylinositol, phosphatidylcholine, and phosphatidylethanolamine) and fatty acids (essential oleic and linoleic acids as well as palmitic acid) are present in the lipid composition of ailanthus seed oil [19]. Other studies on the oil report that it is a complex mixture of saponifiable and unsaponifiable fractions, where the former include fatty acids and triacylglycerols and the latter tocopherols, sterols, and other compounds [3]. It is to the composition that the authors attribute the proven antimicrobial, highly analgesic, antioxidant, and anti-inflammatory activities. Their studies also proved its safety in mouse experiments.

So far, research works on possible applications of the oil of the tree species *A. altissima* are mainly focused on its use in biodiesel production as an environmentally competitive feedstock [20,21,22]. The fuel properties of the biofuel derived from the glyceride oil of ailanthus seeds are highly appreciated and meet the requirements for international standards. All authors point out that isolated glyceride oil has a significant yield to be a cheap and sustainable source for biofuel production. The use of the seeds for biodiesel production is reported for other species of the genus *Ailanthus*, namely *A. excelsa* [23]. The encouraging results on chemical components and biological activities (antimicrobial, antioxidant, and DNA nicking protective activities) that we found for aerial parts of *A. altissima* gave us the reason to continue with the findings on seed oil for healthy food purposes [24,25,26].

The aim of the present study is to determine the content of bioactive lipid-soluble components in *Ailanthus altissima* seed oil, its physicochemical parameters, and fatty acid composition and to verify in vitro its potential for protecting DNA from oxidative damage, its antiproliferative activity, and harmlessness.

## 2. Materials and Methods

### 2.1. Plant Material and Isolation of the Seed Oil

The well-matured seeds from the “tree of heaven” fruits (samaras) were collected in September 2023 from а park area in Plovdiv city (42°08′19″ N, 24°46′59″ E), Bulgaria (Figure 1). Herbarium specimens were deposited in the Agricultural University of Plovdiv Bulgaria’s herbarium, under number No. 063263. The seeds were air-dried to constant weight at room temperature, without direct sunlight. A laboratory mill was used for crushing the seeds with a mesh 30 of the material’s particle size. A Soxhlet extractor was used for the extraction of the oil for 8 h (with a ratio of seed weight to volume of the solvent hexane 1:20) [27]. The extraction yield (seed weight/oil weight ratio) was 3.26:1. After the extraction, the solvent was evaporated in a rotary vacuum evaporator and the rest of the hexane was removed under a stream of nitrogen.

### 2.2. Chemicals, Cell Culture Reagents, and Lines

The chemicals used for determination of the chemical and lipid compositions were supplied by Merck (KGaA, Darmstadt, Germany): n-hexane—CAS No. 110-54-3; methanol—CAS No. 67-56-1; certified reference material Supelco 37 comp. FAME mix—EC No. 200-838-9; 1,4-dioxane—CAS No. 123-91-1; perchloric acid, ACS reagent, 60%—CAS No. 7601-90-3; sulfuric acid, ACS reagent, 95.0–98.0%—CAS No. 7664-93-9; DL-α-, DL-β-, DL-γ-, and DL-δ-tocopherols with purity of 98%. The standard mixture of sterols used were β-sitosterol (with a 10% campesterol, 75% β-sitosterol, Acros Organics, Morris Plains, NJ, USA), cholesterol (stabilized, purity 95%, Acros Organics, Morris Plains, NJ, USA), and stigmasterol (purity 95%, Sigma-Aldrich, St. Louis, MO, USA).

The following chemicals used for the DNA nicking protection test were supplied by Sigma-Aldrich Chemie GmbH (Steinheim am Albuch, Germany): Trolox (6-hydroxy-2,5,7,8-tetramethylchromane-2-carboxylic acid)—CAS No. 238813; potassium phosphate dibasic—CAS No. P3786; hydrogen peroxide solution—CAS No. H1009. From Merck (Darmstadt, Germany), the following chemicals were supplied: di-potassium hydrogen phosphate—CAS No. 1051015000; iron (II) sulfate heptahydrate—CAS No. F7002. And from Duchefa (Haarlem, The Netherlands), we were supplied with TBE buffer—CAS No. T1507—and agarose SPI—CAS No. A1203.

Cell culture reagents and lines for in vitro cytotoxicity and antiproliferative tests were purchased from Sigma-Aldrich, Schnelldorf, Germany: Dulbecco’s modified Eagle’s medium (DMEM), fetal bovine serum (FBS), antibiotics (penicillin and streptomycin), and neutral red; from Orange Scientific, Braine-l’Alleud Belgium: the disposable consumables; and from American Type Cultures Collection (ATCC, Manassas, VA, USA): BALB/c 3T3 clone A31 (ATCC^®^ CСL-163^TM^)—mouse embryonic fibroblast, MCF-10A (ATCC^®^ CRL-10317™)—normal human epithelial, PC3 (ATCC^®^ CRL-1435™)—prostate carcinoma, and HT-29 (ATCC^®^ HTB-38™)—colorectal adenocarcinoma cell lines.

### 2.3. Chemical, Physicochemical, and Chromatographic Methods for the Analysis of the Oil

#### 2.3.1. Chemical Composition

Total protein, moisture, fiber, and ash content were determined according to [28]. Total carbohydrates were calculated by the following formula: 100 − (protein + lipids + water + ash) g/100 g of dry seeds [29]. Water-soluble sugars and the starch content were determined by BS [30,31].

#### 2.3.2. Physicochemical Properties

Standard procedures [32,33,34,35,36,37] were used for the determination of the main physicochemical properties of the seed oil: acid value, peroxide value, saponification value, iodine value, refractive index, and relative density. The oxidative stability of the oil was measured by Rancimat 679 (Metron, Switzerland) at 100 °C [38].

#### 2.3.3. Chromatographic Method for Determination of Fatty Acid Composition

Gas chromatography (GC) was applied for the determination of the fatty acid composition of the oil [39]. Fatty acid methyl esters (FAMEs) were obtained by pre-esterification using methanol with sulfuric acid [40]. Their determination was carried out on a GC unit Agilent 8860 (Santa Clara, CA, USA) with a capillary column (DB-Fast FAMEs) with the following characteristics: 30 m × 0.25 mm × 0.25 μm (film thickness). The detector used was a flame ionization one. The column temperature was 70 °C (for 1 min), up to 250 °C at a rate of 5 °C/min (hold at this temperature for 3 min); the temperature of the injector was 270 °C, and of the detector −300 °C. Identification was through a comparison of the retention times with that of a standard mixture of FAMEs (Supelco, Bellefonte, PA, USA 37 comp. FAMEs mix) run to GC under identical conditions.

#### 2.3.4. Determination of Sterols, Tocopherols, and Phospholipids

The unsaponifiable matter of the oil was determined according to ISO [41]. Total sterols were determined spectrophotometrically using the following procedure. The sterols were isolated by thin-layer chromatography (TLC) from the other unsaponifiables. After that, they were eluted with chloroform and the solvent was evaporated on a rotary vacuum evaporator. Then, to the residue we added 2.8 mL chloroform, 0.4 mL acetic acid, 2 mL 12% sulfosalicylic acid dissolved in acetic acid, 6 mL acetic anhydride, and 0.8 mL sulfuric acid. The sample was kept in a dark place for 20 min in order for the reaction to take place, and after that, the absorbance was immediately measured spectrophotometrically at a wavelength of 597 nm. The blank sample was chloroform [42]. The individual sterols were detected on HP 5890 gas chromatograph with a capillary column DB − 5 (25 m × 0.25 mm) and flame ionization detector. The temperature was 90 °C (3 min) at a rate of 15 °C/min up to 290 °C and then up to 310 °C at a rate of 4 °C/min (10 min). The temperature of the detector was 320 °C and of the injector −300 °C. The carrier gas was hydrogen. Identification was performed by comparison of the retention times with a standard mixture of sterols [43].

Total and individual tocopherols were determined by high-performance liquid chromatography (HPLC). The used equipment was Merck-Hitachi with fluorescent detection (295 nm excitement and 330 nm emission) and column Nucleosil Si 50-5 (250 mm × 4 mm). The mobile phase was hexane/dioxane, 96:4 (*v*/*v*) with a flow rate of 1 mL/min [44].

The phospholipid classes were isolated by two-dimensional TLC [45]. The quantification of the separated phospholipids was performed spectrophotometrically at 700 nm after their mineralization with perchloric and sulfuric acids (1:1, *v*/*v*) [46].

### 2.4. In Vitro Method for DNA Nicking Protection Assay

To assess the DNA protective effect of *A. altissima* seed oil, supercoiled pUC19 plasmid DNA was treated with Fenton’s reagent in the presence of different concentrations of the tested compound as described earlier [47]. Tenfold serial dilutions used in the experiment were prepared in a 12% solution of methylated cyclodextrin (CAVASOL W7 M, Wacker Chemie AG, Munich, Germany) and then added to 450 ng of pUC19. The final reaction volume was adjusted to 20 µL using Milli Q water. The reaction was carried out at 37 °C for 30 min. Ten microliters of Trolox solutions with concentrations of 25, 50, and 100 mg/mL were used as positive reaction controls. Plasmid DNA nicking was visualized on 1.5% agarose gel electrophoresis in 1 × TBE buffer at 120 V for 1.5 h. The DNA damage was semi-quantified using the Gel Doc™ EZ Imaging system (Bio-Rad, Hercules, CA, USA).

### 2.5. In Vitro Tests for Cytotoxicity and Antiproliferative Activities

The in vitro tests showed results in terms of cytotoxicity and antiproliferative activity of the oil on different cell lines. The data obtained were analyzed using mathematical and statistical methods, and the results were visually presented through sigmoidal curves.

#### 2.5.1. Cell Cultivation

Adherent cell lines were grown in plastic cell culture flasks 25 cm^2^ and 75 cm^2^, using DMEM medium (4.5 g/L glucose) supplemented with 10% fetal bovine serum, 100 U/mL penicillin, and 0.1 mg/mL streptomycin. The cells were grown at a temperature of 37 °C with a 5% CO^2^ atmosphere to support their logarithmic growth phase. The cells were trypsinized and then seeded in 96-well plates for cell culture. After 24 h of cultivation under the specified conditions, the cells were treated with the test substance [47].

#### 2.5.2. Determination of Cytotoxicity and Antiproliferative Activities

The NRU assay (neutral red uptake) is a commonly used method in cell biology research to evaluate the cytotoxicity of compounds. It involves staining cells with a dye called neutral red, which is taken up by healthy cells. The intensity of the dye uptake is measured and used as an indicator of cell viability. The test rapidly measures the average cytotoxic concentrations (CC_50_ values) of the substances being researched. This evaluation assists in assessing the substance’s potential toxicity and serves as a starting point for further toxicology experiments, both in vitro and in vivo, utilizing diverse animal models.

Mouse embryonic fibroblasts (BALB/c 3T3, clone А31) were cultured as an adherent, monolayer cell culture in 75 cm^2^ dishes under standard conditions. The cells were initially seeded at a density of 1 × 10^4^ cells/100 μL of culture medium per well in 96-well plates. Following a 24 h incubation for cell adhesion, the cells were treated with a solution of test substances, with concentrations increasing in a double-fold manner. After another 24 h, a culture medium containing neutral red was added, and the cells were incubated for an additional 3 h. Subsequently, the wells were washed with PBS (pH 7.4), and a desorb solution (ethanol/acetic acid/dH_2_O = 49/1/50) was added. The optical density of the samples was measured at λ = 540 nm using a TECAN microplate reader. The cytotoxicity was calculated in percentage [47].

The assay for antiproliferative activity followed a similar procedure but with a slight modification. In this case, the cells were seeded at a lower density of 1 × 10^3^ cells/100 μL culture medium per well. After seeding, the cells were incubated for 72 h with the substance being investigated. The result of the assay was then read.

### 2.6. Statistical Analysis

Physicochemical parameters, compound compositions in the oil, and seed compositions were analyzed three times. Results are expressed as mean ± standard deviation (SD). One-way ANOVA was applied to the obtained data, followed by Tukey’s HSD (honestly significant difference) test and Student’s unpaired *t*-test (*p* < 0.01) [48]. Bonferroni’s post hoc test (Graph Pad Prism 8 Software, San Diego, CA, USA) was used after ANOVA to analyze cytotoxicity and antiproliferative activity results (*n* = 6, from three experiments). Data were presented as mean ± SD (*p* < 0.05).

## 3. Results and Discussion

### 3.1. Chemical Composition of А. altissima Seeds

The isolated *А. altissima* seed oil was liquid with a light yellow color. The results for the content (%) of the oil, proteins, carbohydrates, fiber, ash, and moisture in the seeds are presented in Table 1. As can be seen from the table, the analyzed plant species is characterized by a relatively high content (30.7%) of oil. The amount of carbohydrates prevailed as well (38.9%), of which the fibers were the most represented. The quantity of starch is significantly low, followed by invert sugar and reducing sugars. Protein content was two times lower than the carbohydrate one. Moisture and ash content were 6.0% and 5.7%, respectively.

The oil content of *A. altissima* seeds was almost twice as high as that determined in the studies of El Ayeb-Zakhama et al. [3] (17.32%) and relatively close to that reported by the authors of [20] (38%) and [22] (40%). A lower yield of multiple folds (0.2–2.3%) is reported for *A. altissima* wood oil, which is indicated as suitable for obtaining biodiesel [21]. Our data support the statement of other authors that the species has a high oil yield. The amount of oil extracted is close to or higher than that reported for black cumin, hemp, evening primrose, and milk thistle seed oils, which are known for their medicinal and nutritional value [2]. According to the researchers who worked on *A. altissima* seed oil, it even exceeds the oil yield obtained from some edible and non-edible seeds [20,22]. The content of cellulose and, respectively, fiber, in the studied seeds of *A. altissima* was in lower values than those in wood (46.7%), while that of ash was significantly higher compared to the 0.5% found for this indicator by the authors of [21].

The quantitative data on fiber give us reason to believe that the seeds of *A. аltissima* would be an excellent source of dietary fiber. Various health effects and benefits for the human body have been documented for the latter [49,50,51]. People consuming larger amounts of dietary fiber are exposed to a lower risk of developing gastrointestinal and cardiovascular diseases, obesity, hypertension, diabetes, etc. Supplements containing them improve blood sugar control, and lower blood pressure and serum lipoprotein levels [50]. Through various mechanisms, they limit the development of cancer cells and participate in the prevention of various forms of cancer—colorectal, ovarian, pancreatic, prostate, breast, and head and neck cancer [51]. Fibers have a healing and protective effect against constipation, hemorrhoids, gastroesophageal reflux, duodenal ulcer, hypertension, diverticulitis, and other diseases [50]. They are part of the nutrients in a healthy diet [49].

Another group of macronutrients present in significant amounts in ailanthus seeds and performing a constructive role in the human body is proteins. Incorporating them into the food diet is essential for wellbeing. Their sources are mainly animal-based products (dairy and meat products), but some negative factors in their production (cruelty to animals, pollution of the environment, negative effect of animal food on health, etc.) are reasons to look for their vegan substitutes such as plant-based proteins. The main sources of such plant-based proteins are cereals (6–15%), nuts (18–38%), and seeds (9–30%), groups of which are shown to be superior in protein content to milk (3–5%), and come close to that of meat (23%) [52]. Plant alternatives to animal proteins are preferred in the vegan diet because they contain bioactive molecules such as polyphenols, vitamins, antioxidants, dietary fiber, etc., which are beneficial compounds that have a positive effect on human health.

### 3.2. Physicochemical Properties of А. altissima Seed Oil

The measured values of the physicochemical parameters of the oil are shown in Table 2.

The peroxide value indicates the amount of primary oxidation products in the oils: peroxides and hydroperoxides. The peroxide value of the studied oil from ailanthus seeds indicates that no oxidation processes occurred during its extraction. An indicator of the content of free fatty acids in the oil is the acid value; for the degree of unsaturation, it is the iodine value; and the saponification value reflects the content of ester bonds and free fatty acids. The obtained results for the acid value of the oil are slightly higher than the requirements of [53] for glyceride oils (up to 4 mg KOH/g). More than six times lower is the acid value (0.64 mg KOH/g) obtained by the authors of [20]. According to the iodine value data (129.4 gI_2_/100 g), *A. altissima* seed oil is characterized as a semi-dry oil, the value of which is within the limits of the widely used soybean (124–139 gI_2_/100 g) and sunflower (118–141 gI_2_/100 g) seed oil [53]. The saponification value corresponds to that established for palm oil (190–209 mg KOH/g) [53], *A. altissima* wood oil (206.3 mg KOH/g) [21], and that reported by the authors of [3] for *A. altissima* oil (192.60 mg KOH/g). It can be seen from Table 2 that the iodine values were lower than those found by other authors—132.11 gI_2_/100 g [20]; 136.77 gI_2_/100 g [3]. *A. altissima* wood oil has a lower iodine value (107.2 g I_2_/100 g) compared to seed oil [21]. The refractive index values correspond to those for cottonseed oil (1.458–1.466), grapeseed oil (1.467–1.477), maize oil (1.465–1.468), mustard seed oil (1.461–1.469) [53], as well as sesame seed oil (1.47) [54]. Also, close to these values was the measured relative density, which matches that reported for palm oil [53]. The oxidative stability of ailanthus oil was relatively low (5.0 h) and was similar to that of refined grapeseed oil (3.7 h) [55]. The team of [3] reported similar to our results for refractive index (1.451) and different for the peroxide value (9.4 meqO_2_/kg), which we would attribute to the autoxidation processes that occurred in them. The latter authors have also published the oxidative stability (4.00 h) of the seed oil of *A. altissima* since these values are close to those obtained in the present work.

### 3.3. Lipid Composition of the Seeds and Seed Oil

The content of the determined lipid-soluble compounds (some of which were biologically active) in the seeds and oil from the “tree of heaven” are presented in Table 3.

Based on the percentage of the lipid-soluble components (in the oil), the best represented were the unsaponifiable matter in oil (3.3%) (which included sterols and tocopherols), followed by phospholipids. Calculations for the seeds showed that these components were as follows: unsaponifiable matter (1.0%); phospholipids (0.3%); and sterols (0.2%). Relatively high were the levels of total tocopherols, whose content in the oil was 414 mg/kg and 127 mg/kg in the seeds.

The total content of sterols in *A. altissima* seed oil is close to the values for sterols in widely used vegetable oils such as soybean (0.18–0.41%), cottonseed (0.27–0.64%), and sunflower (0.24–0.46%) [53]. The total tocopherols in the studied oil are similar to those indicated in grapeseed oil (240–410 mg/kg) and safflower oil (240–670 mg/kg) according to [53]. By this indicator, it surpasses some of the most used oils such as walnut oil (209.4 mg/kg), peanut oil (259.6 mg/kg), olive oil (311.0 mg/kg), tomato seed oil (345.8 mg/kg), and evening primrose oil (391.9 mg/kg), and has a similar quantitative value to that reported for *Camellia* oil (416.0 mg/kg) [1]. The higher number of total tocopherols in the oil contributes to its higher resistance to oxidation, which is important to keep its qualities unchanged for a longer time. *A. altissima* seed oil is superior in unsaponifiable matter content to other species of the genus. In *А. Exelsa*, the amount of this group of bioactive components is 2.0% in the refined oil and 1.2% after its refining, respectively [23].

The gas chromatographic analysis of the fatty acid composition of *A. altissima* seed oil showed the presence of eleven fatty acids (Table 4, Figure 2), among which the compositions determined were unsaturated linoleic (48.6%) and oleic (44.8%) acids. The content of saturated palmitic and stearic acids was below 3%, and the other identified fatty acids were present in significantly smaller amounts (0.2–0.6%). The lowest percentage was recorded for saturated margaric acid. The total content of saturated (SFAs), mono- (MUFAs), and polyunsaturated fatty acids (UFAs) in the oil, presented in Table 4, showed that unsaturated fatty acids (UFAs) predominated in *A. altissima* seed oil (95.3%), and a more even distribution of mono- and polyunsaturated fatty acids was observed (46.3% and 49.0%, respectively). Saturated fatty acids in the lipid fraction were only 4.7%.

The total content of saturated and unsaturated fatty acids in ailanthus seed oil was similar to the results obtained by the authors of [3], 4.74 and 95.37%, respectively, but they established more polyunsaturated fatty acids (56.15%). In the present study, the measured amounts of total PUFAs in ailanthus seed oil were similar to corn oil (49.74%), cress oil (48.86%), milk thistle seed oil (48.81%), pumpkin seed oil (48.14%), and jackfruit seed oil (46.72%), while that of MUFAs corresponded to palm oil (46.3%), rice bran oil (43.7%), sesame oil (42.0%), and date palm seed oil (49.59%). The low SFA content corresponds to pomegranate seed oil (5.35%), with values lower even than those reported in various beneficial vegetable oils [1].

The descending order found In the study (linoleic acid > oleic acid > palmitic acid > stearic acid) is also reported in [3], with the best represented fatty acids in amounts similar to ours (55.76% and 38.35% for linoleic acid and oleic acid, respectively). For palmitic and stearic acid, the authors established the same quantities as we found. Other authors’ research confirms the above arrangement, with some variation in percentage content (linoleic (56.2–37.35%), oleic (23.1–25.53%), and palmitic acids (11.1–2.01%)) [19,20]. A similar fatty acid composition, with linoleic acid predominating (50.8%), followed by palmitic acid (30.5%), oleic acid (8.1%), and *α*-linolenic acid (7.3%) is reported by the authors of [21] for oil obtained from the wood of *A. altissima*. Studies on the fatty acid composition of *Ailanthus exelsa* seed oil showed a higher percentage of palmitic (12.6%) and stearic acids (7.6%) [23]. The authors indicated that oleic acid was predominant in the fraction (65.6%) and linoleic acid content was 11.7% (more than four times lower than that found in the present study for *A. altissima*.

Taking into account the fatty acid composition found, we can consider that the species is suitable for filling deficiencies related to the levels of linoleic and oleic acids. As an essential fatty acid, linoleic acid is obtained with food and it is responsible for the proper functioning of the nervous system. It participates in the construction and strength of the cell membrane, the synthesis of some hormones (thyroid and adrenal glands), and has healing properties for human hair and skin [56]. In the review article by the aforementioned authors, healthy oils are commented on, among which pumpkin seed oil is indicated as a source of unsaturated fatty acids (73.1–73.7% total unsaturated fatty acid), in which linoleic acid (39.5–47.5%) dominates, followed by oleic acid (16.6–25.5%). For comparison, the seed kernels of custard apple contain oleic acid 47.4% (close to that found in ailanthus oil) and twice as low linoleic acid (22.9%) [56]. The *A. altissima* seed oil has a similar composition to sesame oil, which is known for its health effects, due to the strong presence of oleic 38.84% and linoleic 46.26% acids [54]. Oils with a high content of monounsaturated fatty acids, in combination with linoleic acid, up to 45% (for better taste), and saturated fatty acids, which give them stability, are suitable for frying [57]. An additional requirement for cooking oils is that they must be liquid, which makes high-oleic oils very suitable (such as sunflower oil, canola oil, and safflower oil). The similar composition of *A. altissima* seed oil suggests similarity in biological activity, nutritional, and other beneficial properties.

### 3.4. Individual Sterol, Tocopherol, and Phospholipid Composition of A. altissima Seed Oil

The good results obtained for the total biologically active components of the oil from *A. altissima* are a reason to deepen our research on its individual sterol, tocopherol, and phospholipid compositions (Table 5). The main component in the sterol composition was *β*-sitosterol (72.6%), followed by stigmasterol (14.0%) and campesterol (10.3%). A certain amount of Δ^7^—stigmasterol (2.1%)—was also found, while the rest of the identified compounds from this group were in significantly smaller amounts (0.3% brassicasterol and cholesterol 0.7%).

El Ayeb-Zakhama et al. [3] also found *β*-sitosterol (70.30%) as the main component of the sterol fraction in ailanthus seed oil. On the other hand, they determined that the second most abundant was campesterol (13.03%), followed by Δ^5^—avenasterol (6.75%) and stigmasterol (6.24%). The individual sterol composition of *A. altissima* seed oil is similar to that of palm kernel oil, where *β*-sitosterol content is the highest (62.6–73.1%), followed by stigmasterol (12.0–16.6%) and campesterol (8.4–12.7%) [53]. Beta-sitosterol is also best represented in wheat germ oil, corn oil, rice bran oil, chili seed oil, and sesame oils (from 400 to 1060 mg/100 g) [58]. The multifunctional role and great pharmacological potential are the reasons for the increased scientific interest in this food component. Pharmacological screening of *β*-sitosterol revealed various antioxidant, anti-inflammatory, anticancer (against breast, prostate, colon, lung, stomach, leukemia, etc.), antimicrobial, immunomodulating, hepatoprotective, and antidiabetic effects. It also helps in the recovery processes of the human body and has a beneficial effect on the cardiovascular system (prevents heart attack and atherosclerosis), and its use as an antihyperlipidemic agent is recommended. It is proven to be harmless, which is a very important characteristic [59,60,61,62]. The second most abundant sterol in the studied oil—stigmasterol—is reported in the same levels in coconut, palm, and sesame oils used for different purposes (food, cosmetic, and pharmaceutical). Campesterol corresponds to amounts measured in coconut, grapeseed, cottonseed, grapeseed, palm, safflower, sunflower, and sesame oils (5–20%) [53]. The presence of stigmasterols in *A. altissima* fruits (ethanol extracts) is associated with the demonstrated antibacterial activity against *Escherichia coli*, *Staphylococcus aureus*, *Pseudomonas aeruginosa*, and *Salmonella typhimurium* [13].

The individual tocopherol composition of the studied oil identified two components: *γ*-tocopherol (74.6%) and *α*-tocopherol, whose content is about three times lower (Table 5). The tocopherol composition of the seed oil from *A. altissima* comes close to that of peanut oil [53] and sesame oil [54], where *γ*-tocopherol is predominant over *α*-tocopherol. According to recent authors, sesame oil is oxidatively stable, which is due to the tocopherols contained in it. Tocopherols are mainly present in vegetable oils and in the form of various isomers, together with tocotrienols, which make up the vital vitamin E. The antioxidant properties of vit. E give reason to regard it as a natural antioxidant that protects the oil from oxidation and preserves its qualities [54,63]. It is an important nutritional component necessary in the human diet, which participates in the disposal of free radicals, the protection of the cell membrane and the prevention of cancer, cardiovascular diseases, Alzheimer’s disease, etc. [63].

From the individual phospholipid composition, four phospholipid classes were identified, the content of which was relatively evenly distributed, varying between 19.8 and 29.5% (Table 5). The highest was the amount of phosphatidylinositol (29.5%), followed by phosphatidic acids and phosphatidylcholine, and the lowest percentage was of phosphatidylethanolamine. Phosphatidic acids were determined for the first time for *A. altissima* seed oil in the present study (25.7%).

A single publication by the authors of [19] reported some phospholipid compounds (phosphatidylinositol, phosphatidylethanolamine, and phosphatidylcholine) in cotyledons and endosperm ailanthus. Of the three phospholipids identified, phosphatidylinositol was equally abundant in endosperm and cotyledons (2.6% in cotyledons, 2.3% in endosperm), while phosphatidylethanolamine predominated in endosperm (3.0%) and phosphatidylcholine in cotyledons (7.0%). The data in the present work are significantly higher than reported.

The phospholipid composition is close to that of linseed oil [1]. Some of the representatives have higher concentrations in ailanthus oil: phosphatidic acids from 2.8 to 25.7 times more, and phosphatidylcholine—1.3–3.6 times more. The percentage content of phosphatidylinositol is the same as that of linseed oil. The phosphatidylcholine and phosphatidylethanolamine present in the oil of ailanthus are the two most common components in eukaryotic cell membranes, with the first occupying the largest share in their composition. The two key macromolecules are found in a certain ratio, the violation of which is associated with the progression of diseases such as diabetes and atherosclerosis [64]. Various biological manifestations are attributed to them, and their presence in food is of particular importance for human health—they exhibit antioxidant properties, improve memory and immunity, and participate in the prevention of cardiovascular diseases [65]. According to recent studies, an important feature of phospholipids is their ability to form complexes with phytoactive compounds or plant extracts. Phospholipid complexation promotes the solubility of the drug, and facilitates its penetration into the body, as well as its sustained release. One of the most important advantages of the formed phospholipid complexes is that they are biocompatible (their main ingredient is usually phosphatidylcholine); therefore, their side effects are weak or absent [66].

### 3.5. In Vitro Cytotoxicity

Cytotoxicity was presented as mean values of CC_50_ (50% cytotoxic concentration) for each tested substance based on the obtained sigmoidal curves (Table 6, Figure 3), and cisplatin is the compound applied as a standard in the present study. In the cytotoxic experiment at an initial concentration of 1 mg/mL of glyceride oil, no effect was recorded. Therefore, the in vitro test continued using higher concentrations in the order of 0.04 to 10 mg/mL on the selected BALB/c 3T3, clone A31 cell line (Figure 3). In the case of cisplatin, the initial cytotoxicity was observed at a concentration from 2.5 to 600 µg/mL. The sigmoidal curves of the graph show significantly increased cytotoxicity of cisplatin over *A. altissima* seed oil.

This is also shown in Table 6—the cytotoxicity exhibited by cisplatin is at a lower concentration of over 173 times, compared to that of the oil tested.

Based on the results obtained, we can summarize that the tested oil from the tree of heaven does not exhibit a cytotoxic effect on mouse embryonic fibroblast, i.e., it is harmless (non-toxic substance).

The lack of cytotoxicity in ailanthus oil has also been reported by previous authors [3]. They indicate the LD_50_ of seed oil at more than 2 g/kg, which proves its potential safe use below this concentration. What they observed is confirmed by the fact that so far no side effects have been reported when using it for treatment. Some studies have focused on investigating the cytotoxicity of ailanthus fruit extracts, but no such data are available for its glyceride oil. The crude extract (methanolic from fruits) and fractions showed good cytotoxic activity attributed to the isolated *β*-sitosterol glucoside [12].

### 3.6. In vitro Antiproliferative Capacity

The antiproliferative activity of the oil was evaluated against the following cell lines: HT-29 (colorectal adenocarcinoma), PC3 (prostate carcinoma), and MCF-10A (normal human epithelial) cell lines. The standard cisplatin that was used for the cytotoxic test was also used for the antiproliferative test. The results of the research are presented in Table 7 and Figure 4.

Different concentrations (from 40 to 10,000 µg/mL) of the oil were tested on all three selected cell lines. No activity was found below 1000 µg/mL concentration for HT-29 and PC3 cell lines. Positive results were observed from 2500 µg/mL to 10,000 µg/mL concentrations for tumor cell lines. The effect and concentrations of oil observed on tumor cell lines are similar to those on the normal human cell line (MCF-10A). An indicator of the ratio that exists between the test substance and the particular cell line (normal or tumor) is the selectivity index (SI). In this case, its value in both tumor lines is low (about 1), indicating that the oil lacks selectivity toward them. The taller than 1.5 values for cisplatin are an expression of greater selectivity.

The present experiment demonstrated a weak antiproliferative effect of ailanthus oil on HT-29 and PC3 cell lines.

The present good antiproliferative results bring new information about the antitumor potential of the species, namely that both its extracts and its seed oil possess such capacity. Until now, antiproliferative activity has been reported only for its extracts (from stem barks, leaves, and roots) and for individual isolated compounds, among which ailanthone is the most significant [6,7]. Ailanthone can be used as an effective anticancer agent, inducing apoptosis in gastrointestinal cancer cells (damages cancer DNA and inhibits its repair [67]. Anticancer effects of *A. altissima* extracts have been described against carcinoma of the thyroid gland, breast, bladder, melanoma, hepatocellular carcinoma, acute myeloid leukemia, human histiocytic lymphoma, osteosarcoma, prostate cancer, lung, liver, etc. [6,7,13]. There are data on potent cytotoxic activity against hepatocarcinoma cells from the leaf extracts, but there is no evidence of similar activity from the oil of the plant species [68].

### 3.7. In Vitro DNA Protective Capacity of the A. altissima Seed Oil

The protective capacity of the *A. altissima* seed oil was assessed in the in vitro DNA nicking assay experiment. To obtain better dispersion of the oil in an aqueous solution, all reactions were set in the presence of 6% methylated cyclodextrin. The results are shown in Figure 5A. Despite the limited extract solubility in aqueous solutions, a clear correlation between the oil concentration and the amount of supercoiled plasmid DNA was observed (Figure 5B). As depicted in Figure 5, the presence of 0.01 μL of the tested oil is sufficient to significantly reduce the amount of nicked DNA. As expected, a similar correlation was found, when different concentrations of Trolox were used as a positive control.

In vitro DNA nicking protection of ethanol leaf, flower, and stem bark extracts of *A. altissima* is reported in our previous study. At concentration of 5.25–10 μg/mL of the extracts a complete protection of plasmid DNA from oxidative damage caused by hydrogen peroxide was found [26].

Other authors also reported a protective potential of ailanthus stem bark extracts on DNA damage caused by zeocin [69]. The latter compared the activity of methanolic and hexane extracts, establishing a higher degree of protection on the tested DNA from yeast cells of *Saccharomyces cerevisiae* of the first extract, whose difference is explained by the better solubility of the bioactive compounds in the hydrophilic agent. The present study adds valuable information on the observed biological activity of another plant product from *A. altissima*, such as its glyceride oil, which has been missing until now.

## 4. Conclusions

The chemical and lipid compositions of the seeds of *Ailanthus altissima*, growing in Bulgaria, have been determined for the first time. High content of glyceride oil, plant-based proteins, and dietary fiber was established. The physicochemical parameters of the seed oil correspond to those of the preferred edible oils, such as soybean, sunflower, palm, grapeseed, etc. The current study confirms the prevalence of unsaturated fatty acids in the oil, with the main components being linoleic and oleic acids. Among the lipid-soluble components identified, *β*-sitosterol, *γ*-tocopherol, phosphatidylinositol, and phosphatidic acids were major; the latter ones have been reported for the first time in the individual phospholipid composition of ailanthus seed oil. The proven high DNA protective ability against oxidative damage, and weak antitumor activity of *A. altissima* seed oil, previously unreported, enrich the available information on the species. Given the composition of fatty acids found, the lipid-soluble bioactive compounds, and the protection against oxidative DNA damage of the studied oil and its lack of toxicity (its harmlessness), it would be an excellent source for inclusion in a healthy diet, in nutritional supplements, in cosmetic products with beneficial effects such as anti-ageing, etc.

## Figures and Tables

**Figure 1 foods-13-01268-f001:**
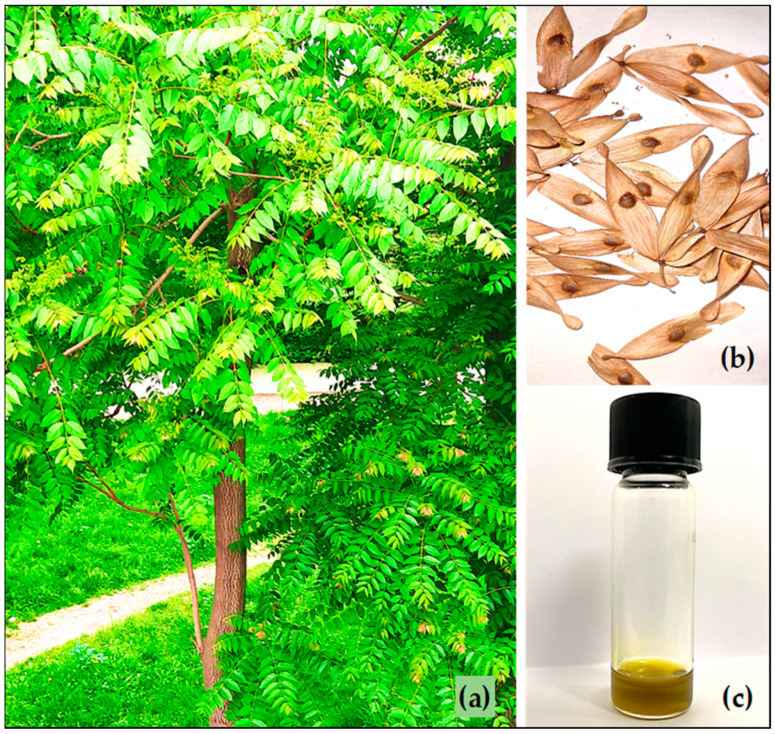
*Ailanthus altissima* tree (**a**), fruits (**b**), and isolated seed oil (**c**).

**Figure 2 foods-13-01268-f002:**
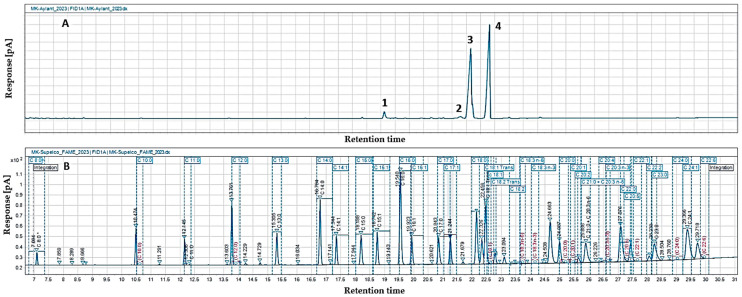
Chromatogram of the main identified fatty acids in the seed oil from *A. altissima* (**A**) and the standard solution of fatty acid methyl esters (FAMEs) (**B**). 1—Palmitic acid. 2—Stearic acid. 3—Oleic acid. 4—Linoleic acid.

**Figure 3 foods-13-01268-f003:**
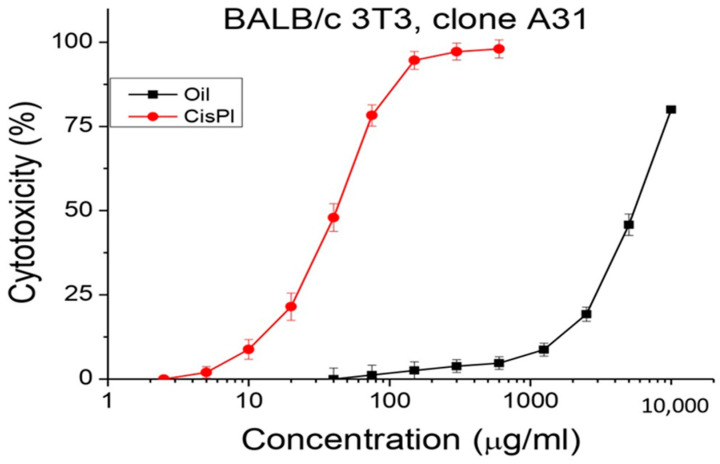
In vitro cytotoxicity test of *A. altissima* seed oil determined in the cell line BALB/c 3T3, clone A31, *n* = 6.

**Figure 4 foods-13-01268-f004:**
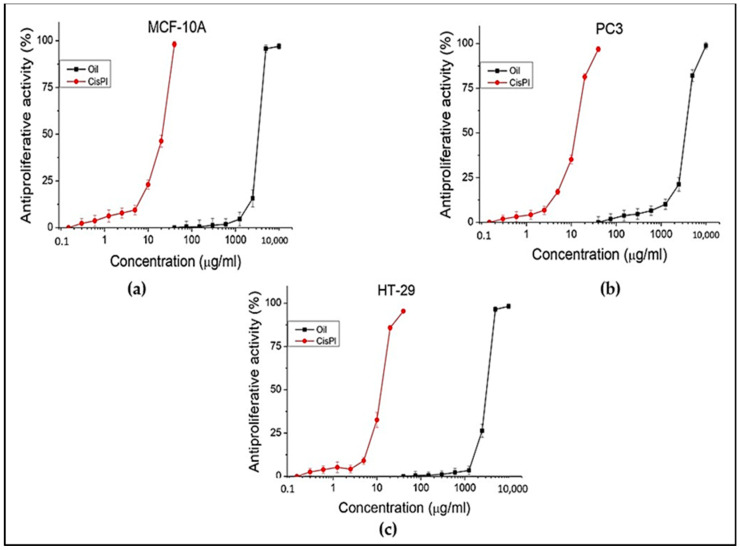
In vitro antiproliferative capacity (in %) of *A. altissima* seed oil on MCFA-10A normal human (**a**), and PC3, and HT29 tumor (**b**,**c**) cell lines, *n* = 6.

**Figure 5 foods-13-01268-f005:**
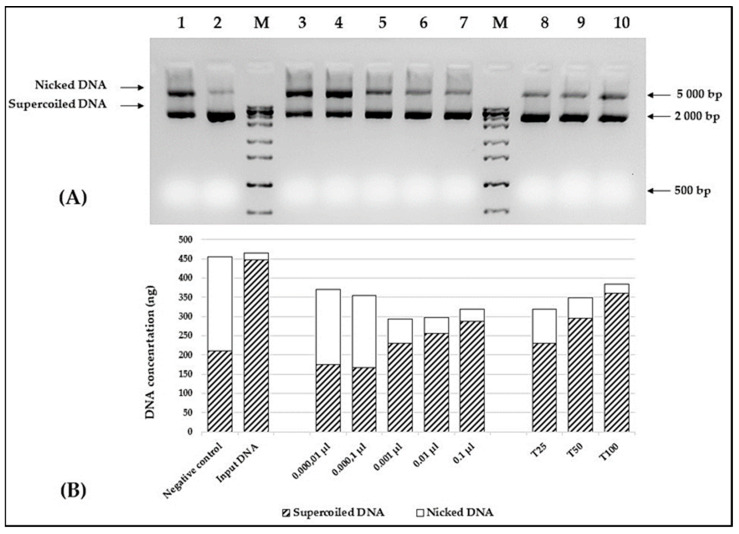
DNA protective activity of *A. altissima* seed oil. (**A**) Agarose gel electrophoresis and (**B**) relative concentration of necked plasmid DNA: line 1—negative control; line 2—plasmid DNA input; lines 3–7—tenfold dilutions of tested oil (0.000,01; 0.000,1; 0.001; 0.01; and 0.1 μL/reaction); lines 8–10—different concentrations of Trolox (25, 50, and 100 mg/mL). 6-ZipRuler 1 Express DNA Ladder (Thermo Scientific, Waltham, MA, USA, cat No SM1373).

**Table 1 foods-13-01268-t001:** Chemical composition of *Ailanthus altissima* seeds.

Parameter	Content (%) *
Oil content	30.7 ± 0.2
Proteins	18.7 ± 0.1
Carbohydrates	38.9 ± 0.6
» starch	4.5 ± 0.1
» reducing sugars	1.3 ± 0.1
» invert sugar	1.7 ± 0.1
» fiber	29.6 ± 0.3
Ash	5.7 ± 0.2
Moisture	6.0 ± 0.1

* The samples were analyzed in triplicate, and the results were expressed as mean ± standard deviation. »—The components noted are part of the total carbohydrates.

**Table 2 foods-13-01268-t002:** Physicochemical properties of the seed oil.

Parameter	Content *
Peroxide value, meq O_2_/kg	0.0 ± 0.0
Acid value, mg KOH/g	4.3 ± 0.1
Iodine value, gI_2_/100 g	129.4 ± 0.3
Saponification value, mg KOH/g	210 ± 2
Relative density	0.8891 ± 0.0002
Refractive index	1.4736 ± 0.0001
Oxidative stability, h	5.0 ± 0.0

* Samples were analyzed in triplicate, and results are expressed as mean ± standard deviation.

**Table 3 foods-13-01268-t003:** Lipid-soluble components in *Ailanthus altissima* seeds and seed oil.

Lipid-Soluble Components	Content
Unsaponifiable matter, %	
-in the oil	3.3 ± 0.2 ^a^
-in the seeds	1.0 ± 0.1 ^b^
Phospholipids, %	
-in the oil	0.9 ± 0.1 ^a^
-in the seeds	0.3 ± 0.0 ^b^
**Sterols, %**	
-in the oil	0.6 ± 0.1 ^a^
-in the seeds	0.2 ± 0.0 ^b^
Tocopherols, mg/kg	
-in the oil	414 ± 12 ^a^
-in the seeds	127 ± 4 ^b^

Samples were analyzed in triplicate, and results are expressed as mean ± standard deviation. Based on the Student’s *t*-test, different superscripts indicate statistically significant differences at *p* < 0.01.

**Table 4 foods-13-01268-t004:** Fatty acid composition of the seed oil.

Fatty Acids		Content * (%)
С_8:0_	Caprylic acid	0.3 ± 0.0
С_15:1_	Pentadecenoic acid	0.3 ± 0.1
С_16:0_	Palmitic acid	2.9 ± 0.1
С_16:1_	Palmitoleic acid	0.3 ± 0.0
С_17:0_	Margaric acid	0.2 ± 0.0
С_17:1_	Heptadecenoic acid	0.3 ± 0.0
С_18:0_	Stearic acid	1.3 ± 0.1
С_18:1_	Oleic acid	44.8 ± 0.4
С_18:2(n−6)_	Linoleic acid	48.6 ± 0.5
С_18:3(n−3)_	*α*-Linolenic acid	0.4 ± 0.1
С_20:1_	Eicosenoic acid	0.6 ± 0.0
Total SFAs		4.7
Total MUFAs		46.3
Total PUFAs		49.0

SFAs—saturated fatty acids; MUFAs—monounsaturated fatty acids; PUFAs—polyunsaturated fatty acids. * Samples were analyzed in triplicate, and results are expressed as mean ± standard deviation.

**Table 5 foods-13-01268-t005:** Individual sterol, tocopherol, and phospholipid composition of the seed oil of *A. altissima*.

Sterols, %		Tocopherols, %		Phospholipids, %	
Cholesterol	0.7 ± 0.1 ^e^	*α*-Tocopherol	25.4 ± 0.2 ^b^	Phosphatidylinositol	29.5 ± 0.1 ^a^
Brassicasterol	0.3 ± 0.0 ^e^	*γ*-Tocopherol	74.6 ± 0.2 ^a^	Phosphatidic acids	25.7 ± 0.2 ^b^
Campesterol	10.3 ± 0.1 ^c^			Phosphatidylcholine	25.0 ± 0.2 ^c^
Stigmasterol	14.0 ± 0.2 ^b^			Phosphatidylethanolamine	19.8 ± 0.1 ^d^
*β*-Sitosterol	72.6 ± 0.4 ^a^				
Δ^7^-Stigmasterol	2.1 ± 0.1 ^d^				

Samples were analyzed in triplicate, and results are expressed as mean ± standard deviation. Based on Student’s *t*-test and Tukey’s HSD test, different superscripts indicate statistically significant differences at *p* < 0.01.

**Table 6 foods-13-01268-t006:** Cytotoxicity (mean CC_50_ values) on BALB/c 3T3.

Compounds	Mean CC_50_ ± SD (µg/mL)
Oil	5432.28 ± 314.64
Cisplatin	31.35 ± 2.72

Results were calculated from three measurements and expressed as mean ± SD.

**Table 7 foods-13-01268-t007:** Antiproliferative activity, values of IC_50_.

Compounds	Mean IC_50_ ± SD (μg/mL)			SI *	
	MCF-10A	PC3	HT-29	PC3	HT-29
Oil	3366.3 ± 95.4	3472.7 ± 107.9	3158.8 ± 94.6	0.97	1.07
Cisplatin	15.7 ± 0.6	9.4 ± 0.3	9.4 ± 0.4	1.67	1.67

* Selective index (SI)—IC_50_ determined by following 72 h treatment with glyceride oil. Results were calculated from three measurements and expressed as mean ± SD.

## Data Availability

The original contributions presented in the study are included in the article, further inquiries can be directed to the corresponding authors.

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
