# Peer review of "Ailanthus altissima Seed Oil—A Valuable Source of Lipid-Soluble Components with DNA Protective and Antiproliferative Activities"

_foods, 2024, doi:10.3390/foods13081268_

Round 1

Reviewer 1 Report

Comments and Suggestions for Authors

The present research describes the extraction of Ailanthus altissima oil, its chemical composition, its anti-proliferative capacity on various types of cancer cell lines as well as its DNA protective capacity.  The manuscript is clearly described, concise and well structured. The experimental protocol is congruent with the objectives sought. In my opinion, this research contains interesting and useful information that meets the quality and novelty requirements of Foods journal. Prior to this, the authors should consider the following observations/suggestions.

Line 52: Write Ailanthus in italics

Section 2.1:

Did the seed dry out? If yes, describe the drying conditions.

Was the seed crushed?

Ratio of seed weight to volume of hexane?

What was the extraction yield (seed weight/oil weight ratio)?

Line 154: Why was a wavelength of 597 nm used to determine total sterol content? Do they absorb in the visible region?

Line 228: Delete “a” before “Table 1”

Lines 313, 323, 360: Are these negative numbers?

Line 310: It seems to me very ambiguous or imprecise to mention "the percentage of bioactive components". That is, how were the bioactive components separated from the non-bioactive components? Or are they all bioactive components?

It will be very useful for the authors to integrate the proportion of mono-, di- and tri-glycerides present in A. altissima oil.

It is recommended to integrate in Figure 1 the chromatogram of the FAME standards, together with the chromatogram shown in the same figure.

Author Response

Dear reviewer,

We send you the revised version of manuscript ID foods-2946410 Ailanthus altissima seed oil - a valuable source of lipid-soluble components with DNA protective and antiproliferative activities”.

We have addressed all the comments raised in the evaluation process and have made the necessary corrections. Some technical mistakes in the text were also corrected. Our responses are given point-by-point after each of the comments. All implemented changes are highlighted by the “Track changes” function in the revised manuscript.

We thank the anonymous reviewers for the valuable comments and suggestions and for the opportunity to improve and re-submit our manuscript.

RESPONSE TO REVIEWER'S COMMENTS

Reviewer 1

R1: The present research describes the extraction of Ailanthus altissima oil, its chemical composition, its anti-proliferative capacity on various types of cancer cell lines as well as its DNA protective capacity. The manuscript is clearly described, concise and well-structured. The experimental protocol is congruent with the objectives sought. In my opinion, this research contains interesting and useful information that meets the quality and novelty requirements of Foods journal. Prior to this, the authors should consider the following observations/suggestions.

R1: Line 52: Write Ailanthus in italics

A: It was corrected in the manuscript.

Section 2.1:

R1: Did the seed dry out? If yes, describe the drying conditions.

A: The seeds were air-dried to constant weight at room temperature, without direct sunlight. (added in text).

R1: Was the seed crushed?

A: A laboratory mill was used for crushing the seeds with a mesh 30 of the material’s particle size. This was added to the manuscript.

R1: Ratio of seed weight to volume of hexane?

A: The ratio of seed weight to volume of the solvent hexane was 1:20. This was added in the manuscript.

R1: What was the extraction yield (seed weight/oil weight ratio)?

A: The extraction yield (seed weight/oil weight ratio) was 3.26:1.

R1: Line 154: Why was a wavelength of 597 nm used to determine total sterol content? Do they absorb in the visible region?

A: Total sterols were determined spectrophotometrically using the following procedure. The sterols were isolated by thin-layer chromatography (TLC) from the other unsaponifiables. After that, they were eluted with chloroform and the solvent was evaporated on a rota-ry-vacuum evaporator. Then, to the residue was added 2.8 mL chloroform, 0.4 mL acetic acid, 2 mL 12% sulfosalicylic acid dissolved in acetic acid, 6 mL acetic anhydride and 0.8 mL sulfuric acid. The sample was kept in a dark place for 20 min in order the reaction to take place and after that the absorbance was immediately measured spectrophotometrically at a wavelength of 597 nm. The blank sample was chloroform.

This explanation was added to the manuscript.

R1: Line 228: Delete “a” before “Table 1”

A: Corrected as suggested.

R1: Lines 313, 323, 360: Are these negative numbers?

A: These are not negative numbers. For that reason, we corrected these parts as suggested.

R1: Line 310: It seems to me very ambiguous or imprecise to mention "the percentage of bioactive components". That is, how were the bioactive components separated from the non-bioactive components? Or are they all bioactive components?

A: The reviewer made the right point – not all of the given components are biologically active. For that reason, we corrected these paragraphs as suggested.

R1: It will be very useful for the authors to integrate the proportion of mono-, di- and tri-glycerides present in A. altissima oil.

A: The reviewer is right that the article will be more detailed when integrating the proportion of mono-, di- and triglycerides present in A. altissima oil. Unfortunately, this determination was not performed and we are not able to incorporate such results in the final manuscript.

R1: It is recommended to integrate in Figure 1 the chromatogram of the FAME standards, together with the chromatogram shown in the same figure.

A: Corrected as suggested.

Reviewer 2 Report

Comments and Suggestions for Authors

Perhaps, the manuscript suffers many formatting errors that could be caused by the careless.

1. The edible history or evidence of Ailanthus altissima, in particular of its seed must be well addressed. In my view, if Ailanthus altissima seed is not a food, the manuscript could be beyond the scope of FOODS.

2. Please distinguish between “seed oil” and “glyceride oil”, the two are confused in many expressions including headlines. As I understand it, the Ailanthus altissima seed oil is a kind of glyceride-rich oil, and in many determinations, the authors measured the seed oil isolated from Ailanthus altissima, rather than glyceride oil.

3. The preparation of Ailanthus altissima seed oil should be detailed. More importantly, is the residue of hexane determined or controlled? Excessive level of hexane is prohibited in food products.

4. The resolution of Fig. 1 must be improved.

5. The figure of chromatogram of the main identified fatty acids should be marked as Fig. 2, its resolution should be improved.

6. Other figures should be polished to make them more vivid, especially the statistical differences should be marked.

Author Response

Dear reviewer,

We send you the revised version of manuscript ID foods-2946410 Ailanthus altissima seed oil - a valuable source of lipid-soluble components with DNA protective and antiproliferative activities”.

We have addressed all the comments raised in the evaluation process and have made the necessary corrections. Some technical mistakes in the text were also corrected. Our responses are given point-by-point after each of the comments. All implemented changes are highlighted by the “Track changes” function in the revised manuscript.

We thank you for the valuable comments and suggestions and for the opportunity to improve and re-submit our manuscript.

RESPONSE TO REVIEWER'S COMMENTS

Reviewer 2

Perhaps, the manuscript suffers many formatting errors that could be caused by the careless.

 R1: 1. The edible history or evidence of Ailanthus altissima, in particular of its seed must be well addressed. In my view, if Ailanthus altissima seed is not a food, the manuscript could be beyond the scope of FOODS.

A: Ailanthus altissima seeds are not traditionally used for food, but the current analysis of the oil composition obtained from them shows that they have very good characteristics, bringing them closer to other well-known and used edible oils. Moreover, the toxicity analysis showed its harmlessness. This gives reason to continue research on the oil for its nutritional use. For this reason, it was submitted for publication in "Foods" Journal.

R1: 2. Please distinguish between “seed oil” and “glyceride oil”, the two are confused in many expressions including headlines. As I understand it, the Ailanthus altissima seed oil is a kind of glyceride-rich oil, and in many determinations, the authors measured the seed oil isolated from Ailanthus altissima, rather than glyceride oil.

A: Corrected as suggested throughout the whole manuscript.

R1: 3. The preparation of Ailanthus altissima seed oil should be detailed. More importantly, is the residue of hexane determined or controlled? Excessive level of hexane is prohibited in food products.

A: Details of the isolation of the oil were given in the manuscript – in 2.1. Plant Material and Isolation of the Seed Oil: “A laboratory mill was used for crushing the seeds with a mesh 30 of the material’s particle size. A Soxhlet extractor was used for the extraction of the oil for 8 h (with a ratio of seed weight to volume of the solvent hexane 1:20) [27]. The extraction yield (seed weight/oil weight ratio) was 3.26:1. After the extraction the solvent was evaporated in a rotary vacuum evaporator and the rest of the solvent was removed under a stream of nitrogen.”

The residue of the solvent after evaporation was not more than 2 mg/100 mL oil which follows the ISO standard (ISO 659:2014; Oilseeds. Determination of Oil Content (Reference Method), ISO: Geneva, Switzerland, 2014).

R1: 4. The resolution of Fig. 1 must be improved.

A: It is done. The figure is improved.

R1: 5. The figure of chromatogram of the main identified fatty acids should be marked as Fig. 2, its resolution should be improved.

A: Corrected as suggested.

R1: 6. Other figures should be polished to make them more vivid, especially the statistical differences should be marked.

A: Corrected as suggested.

Reviewer 3 Report

Comments and Suggestions for Authors

In general, the manuscript is written correctly and has adequate scientific support. The authors in this paper have addressed the phytochemical study, including the content of liposoluble compounds in the oil obtained from the seeds of Ailanthus altissima (Simaroubaceae). Cytotoxicity, antiproliferative activity and DNA protective capacity of glyceride oil were also examined .

Comments on the Quality of English Language

Minor editing required.

Author Response

Dear reviewer,

We send you the revised version of manuscript ID foods-2946410 Ailanthus altissima seed oil - a valuable source of lipid-soluble components with DNA protective and antiproliferative activities”.

We have addressed all the comments raised in the evaluation process and have made the necessary corrections. Some technical mistakes in the text were also corrected. Our responses are given point-by-point after each of the comments. All implemented changes are highlighted by the “Track changes” function in the revised manuscript.

We thank you for the valuable comments and suggestions and for the opportunity to improve and re-submit our manuscript.

RESPONSE TO REVIEWER'S COMMENTS

Reviewer 3

R1: In general, the manuscript is written correctly and has adequate scientific support. The authors in this paper have addressed the phytochemical study, including the content of liposoluble compounds in the oil obtained from the seeds of Ailanthus altissima (Simaroubaceae). Cytotoxicity, antiproliferative activity and DNA protective capacity of glyceride oil were also examined.

A: We are grateful to the reviewer for the given suggestion for improving the quality of the manuscript.

R1: Comments on the Quality of English Language

Minor editing required.

A: Additionally, the detected technical and English language mistakes were corrected in the revised manuscript.

Round 2

Reviewer 2 Report

Comments and Suggestions for Authors

After revision, the manuscript is ready for acceptance.